# Phasor-Based Myoelectric Synergy Features: A Fast Hand-Crafted Feature Extraction Scheme for Boosting Performance in Gait Phase Recognition

**DOI:** 10.3390/s24175828

**Published:** 2024-09-08

**Authors:** Andrea Tigrini, Rami Mobarak, Alessandro Mengarelli, Rami N. Khushaba, Ali H. Al-Timemy, Federica Verdini, Ennio Gambi, Sandro Fioretti, Laura Burattini

**Affiliations:** 1Department of Information Engineering, Università Politecnica delle Marche, 60131 Ancona, Italy; r.mobarak@pm.univpm.it (R.M.); a.mengarelli@staff.univpm.it (A.M.); f.verdini@staff.univpm.it (F.V.); e.gambi@staff.univpm.it (E.G.); s.fioretti@staff.univpm.it (S.F.); l.burattini@staff.univpm.it (L.B.); 2Transport for NSW Alexandria, Haymarket, NSW 2008, Australia; rkhushab@gmail.com; 3Biomedical Engineering Department, Al-Khwarizmi College of Engineering, University of Baghdad, Baghdad 10066, Iraq; ali.altimemy@kecbu.uobaghdad.edu.iq

**Keywords:** gait, EMG, myoelectric control, feature extraction, deep learning, lower limb, assistive devices

## Abstract

Gait phase recognition systems based on surface electromyographic signals (EMGs) are crucial for developing advanced myoelectric control schemes that enhance the interaction between humans and lower limb assistive devices. However, machine learning models used in this context, such as Linear Discriminant Analysis (LDA) and Support Vector Machine (SVM), typically experience performance degradation when modeling the gait cycle with more than just stance and swing phases. This study introduces a generalized phasor-based feature extraction approach (PHASOR) that captures spatial myoelectric features to improve the performance of LDA and SVM in gait phase recognition. A publicly available dataset of 40 subjects was used to evaluate PHASOR against state-of-the-art feature sets in a five-phase gait recognition problem. Additionally, fully data-driven deep learning architectures, such as Rocket and Mini-Rocket, were included for comparison. The separability index (SI) and mean semi-principal axis (MSA) analyses showed mean SI and MSA metrics of 7.7 and 0.5, respectively, indicating the proposed approach’s ability to effectively decode gait phases through EMG activity. The SVM classifier demonstrated the highest accuracy of 82% using a five-fold leave-one-trial-out testing approach, outperforming Rocket and Mini-Rocket. This study confirms that in gait phase recognition based on EMG signals, novel and efficient muscle synergy information feature extraction schemes, such as PHASOR, can compete with deep learning approaches that require greater processing time for feature extraction and classification.

## 1. Introduction

In the last two decades, there has been an increasing trend in the development of specific sensors and technologies for human–machine interfaces (HMIs) [1]. This trend was undoubtedly driven by the availability of minimally intrusive electromyographic (EMG) probes, which allowed recording large sets of information related to the human motion of upper and lower limbs in an easy way [2,3,4,5]. Additionally, with the increased computational power in embedded systems, these sensors eventually enabled the applicability of signal processing and machine learning techniques capable of decoding the volitional intent of human beings. This aspect was crucial for implementing smart interactions of the limbs with active prostheses and assistive devices [6,7,8], making the use of such technologies particularly advantageous in motor rehabilitation scenarios [9,10]. However, it is worth noting that the majority of myoelectric technology solutions presented in the literature have been developed with a primary focus on the upper limb. Efforts to integrate EMG-based HMIs into the control policies of lower limb prostheses have been relatively marginal compared to the upper limb and have only recently received specific attention [11,12,13].

As emerged in [11,13], lower limb prostheses and assistive exoskeletons fall into three main categories: passive, semi-active, and active. The first category of devices is entirely mechanical, whereas semi-active and active systems incorporate microprocessor systems that use mechanical information to control artificial joint impedance or provide propulsion, often through actuators or motors that eventually compensate for lost musculature [14]. In any case, the commercially available systems do not exploit bioelectronic signals like EMG [12], even if leveraging neuromuscular information in semi-active and active prostheses can enhance the user experience, making it more biomimetic, functional, and superior [5,11]. A question that arises concerns why myoelectric control for artificial lower limb systems is underutilized despite its benefits [11,13]. To find an answer, one has to consider the nature of the EMG signal [12]. Indeed, although reflecting limb movement intention, it is a highly stochastic signal and may be particularly sensitive to electrode shifts or muscle adaptation due to the rise in fatigue [12,15]. Moreover, misinterpretation by the myoelectric control system can lead to improper human–machine interaction, potentially endangering the user by causing unexpected falls [13]. This imposes that the high-level control subsystem of the HMI has to be particularly robust to finely detect transitions of the gait phase and locomotion modalities [13,16]. Thus, although the use of proportional myoelectric control in the higher-level controller of such devices represents a final goal for HMIs, EMG-based pattern recognition solutions still appear appealing to be investigated and transferred in embedded systems since they provided robust performance in different control solutions [11,12,13,17].

In this context, EMG data are mapped to specific gait phases of the gait cycle by the trained pattern recognition system [17,18,19]. However, no consensus in the literature has been reached regarding the granularity of the gait cycle [20], i.e., how many phases of the gait cycle one considers and the modalities in which the latter is partitioned. Recent studies suggest partitioning the gait cycle with a granularity of five phases, i.e., five classes for the myoelectric pattern recognition architecture as in [21,22]. Although eight phases can be recognized in the gait cycle [20], a partitioning into five can represent a good compromise for ensuring smooth motion transitions and reliable gait phase recognition of the lower limb assistive device [21]. Moreover, it is interesting to notice that standard myoelectric pattern recognition architectures employed in discriminating among five phases of the gait cycle show a mean accuracy of around 85% with great variability among the subjects [21,22,23]. These results obtained with consolidated myoelectric pattern recognition methodologies open up possibilities for further investigations. Among the various aspects worth mentioning, two play a central role in the development of next-generation myoelectric pattern recognition systems for the lower limb [11]. First, it is essential to understand whether the use of recent deep learning architectures used for upper limb applications is indeed advantageous in the case of the lower limb [24]. Second, one should explore whether lightweight machine learning architectures commonly employed in the state-of-the-art can be improved by applying novel feature extraction methodologies.

Regarding the first point, there is a notable trend towards using deep learning for hand gesture recognition aimed at myoelectric control of upper artificial limbs [25,26], which deserves investigation for lower-limb applications. Architectures such as Convolutional Neural Networks (CNNs) have assumed a central role in this context [26,27]. However, they require tuning a large number of parameters, which requires high computational expense during the training phase and can occupy hundreds of megabytes of memory during the model storage phase [27,28], which can be a non-optimal approach when dealing with embedded systems that have to allocate memory for managing multiple processes before passing the decision to the lower-level controller. Additionally, both training and operational phases require high volumes of data [25,27]. Such volumes can be obtained through high-density EMG sensors surrounding the area of interest on the upper limb or through armbands containing at least eight electrodes [29,30]. This is not always possible for the lower limb, where the literature shows a greater use of sparse surface EMG probes due to the physical conformation of the leg and the different muscles involved in gait [4,31,32]. It is therefore essential to understand whether, with sparse setups usable on the lower limb, deep learning techniques can efficiently solve the problem of gait phase detection, thus understanding whether benefits in terms of reliability and accuracy merit the cost of the high computational power and memory required to maintain deep learning in the control system. The aforementioned considerations are closely related to the second aspect previously mentioned, i.e., improving the performance of pattern recognition architectures such as Support Vector Machine (SVM) and Linear Discriminant Analysis (LDA), which have proven robust in myoelectric control problems for both upper and lower limbs [21,30,33,34], through the use of next-generation feature extraction algorithms. Indeed, it is notable that CNN-based approaches can enhance accuracy performance through convolutional blocks, which capture the spatial relationships between muscle activation patterns of different input EMG signals [27]. In contrast, LDA and SVM are generally used with hand-crafted features [21,30], which do not prioritize the extraction of information related to the spatial synergies of muscle activation. A possibility is given by extracting synergistic features by using non-negative factorization approaches [35,36,37]. However, such approaches require learning a static map from muscle signals to muscle synergy space expressed in the form of a matrix, the coefficients of which may require re-calibration as it happens in many myoelectric control schemes that employ feature reduction approaches [38].

Hence, the aim of this study is to develop and assess the reliability of a modern feature extraction approach able to embed spatial muscle synergy information in the feature space that can be useful to upgrade the performance of consolidated HMIs for lower-limb myoelectric control applications. To do this, the concept of spatial information embedding through the phasor approach (PHASOR) was used [39]. This avoids the need for training specific maps through factorization methods to include spatial synergistic information. To demonstrate the benefits of PHASOR sets in myoelectric-based gait phase detection, a detailed comparison with other state-of-the-art EMG feature sets was carried out. Moreover, following the rationale of the study, a deep learning approach using convolutional kernels was included in the comparison. In particular, the selected architecture has recently been presented in the field of deep learning time series classification and is known as Random Convolutional Kernel Transform (Rocket), and the faster variant is the Minimally Random Convolutional Kernel Transform (Mini-Rocket) [40,41]. Such approaches were included since they showed great classification accuracy with many different time series but, differently from CNN, showed extremely reduced training times since they use a single layer of a large variety of randomly initiated convolutional kernels that do not require training of the coefficients [40]. Analogous to how the Extreme Learning Machine (ELM) is to Artificial Neural Networks (ANNs), Rocket can be considered similar to Convolutional Neural Networks (CNNs). However, while ELM has been widely used for myoelectric control [42], Rocket and Mini-Rocket appear to have been less employed for this purpose, being only recently applied in the classification of wrist movements [24]. In this study, Rocket will therefore be investigated in the context of gait phase detection as a benchmark for comparison with the PHASOR approach.

## 2. Materials and Methods

### 2.1. Dataset Presentation and Pre-Processing

The data employed in this study belong to a publicly available repository called SIAT-LLMD [21], which contains a detailed collection of EMG signals and kinematic data specifically tailored to capture lower limb movements, including typical walking patterns [21]. A total of 40 subjects, 30 male and 10 female, were considered in this study. Raw EMG signals were collected using a surface electromyographic recording system, with a sampling frequency of 1920 Hz [21]. Probes were placed to record the myoelectric activity of 9 leg muscles of the left side, i.e., tensor fascia lata, rectus femoris, vastus medialis, semimembranosus, upper tibialis anterior, lower tibialis anterior, lateral gastrocnemius, medial gastrocnemius, and soleus. A stereophotogrammetric system, synchronous with EMG recording systems, recorded the kinematics of infrared reflective markers, while ground reaction forces were collected through two force-plates [21]. Further details regarding the walking locomotion modes and the processed variables obtained and gathered in the repository can be found in [21].

For this study, only free-walking condition data were considered. A total of 10 gait cycles per subject were provided by the authors of the dataset, along with the labeled class vectors for the phases of the gait cycles. The authors adopted a granularity level of five classes for partitioning the gait cycles [20,21], namely, heel strike (HS), flat foot (FF), mid-stance (MS), heel-off (HO), and toe-off (TO). These vectors constitute the ground truth for training and testing the pattern recognition models, thereby ensuring the reliability of comparisons between the developed pattern recognition systems [21]. Hence, raw EMG data from each subject for each gait cycle were pre-processed using a zero-lag Butterworth band-pass filter of the fourth order, with a frequency range between 10 and 400 Hz [43]. The filtered data were then used in specific feature extraction pipelines as defined in the following section.

### 2.2. PHASOR-Based Features and State-of-the-Art Feature Sets

The PHASOR feature extraction approach maps synergistic muscle activation patterns through a complex representation, specifically polar coordinates. This method has shown particular benefits in upper-limb myoelectric control problems, where the electrodes are placed radially around the circumference of the residual part of the limb [39]. The PHASOR transformation can potentially be applied to any feature computed from a set of radial electrodes, as it requires associating each electrode with a unique position represented by a polar coordinate [39]. An intuitive way to achieve this is by partitioning the 2π angle into a set of k2π/N positions, where k is the index of the channel that ranges from 0 to N − 1, and *N* is the total number of channels present in the recording system. This approach allows for the creation of a multidimensional feature expressed by an *N*-dimensional vector, where each element is given by
(1)$$f_{k}e^{j\frac{k2\pi }{N}}\;\;\mathrm{with}\;k=0\ldots N-1$$
where $f_{k}$ is the *f* feature type, i.e., waveform length (WL), root-mean-square (RMS), and so on, extracted from the *k*th channel. Hence, the PHASOR representation of the feature $P^{f}$ characterized with respect to the nine EMG channels used in this study can be written as follows:(2)$$P^{f}=\left[\begin{array}{ccccc} f_{0} & f_{1}e^{j\frac{2\pi }{9}} & f_{2}e^{j\frac{4\pi }{9}} & \cdots & f_{8}e^{j\frac{16\pi }{9}} \end{array}\right]$$

The PHASOR representation is then used to highlight the spatial synergistic information of muscle activation patterns in the new feature set by computing the modulus of the difference between each single element of $P^{f}$ [39]. This results in feature vectors of dimension N·(N−1)/2 that can be represented by $D_{f}$, with elements given by
(3)$$|f_{k}e^{j\frac{k2\pi }{N}}-f_{l}e^{j\frac{l2\pi }{N}}|\;\;\mathrm{with}\;k\neq l$$

Modifying the approach suggested in [39], one can generalize an f-PHASOR feature set of the following form:(4)$$f\mathrm{-PHASOR}=\left[\begin{array}{cc} \log(D_{f}) & \log(D_{f}/\nabla D_{f}) \end{array}\right]$$
where $\nabla D_{f}$ is obtained using the same methodology over the time derivative of the EMG signals estimated through simple numerical differentiation.

In this study, the *f*-PHASOR approach was used with waveform length (WL) and Root Mean Square (RMS), resulting in WL-PHASOR and RMS-PHASOR. The equations for computing WL and RMS, given a sliding window of L samples, are as follows:(5)$$\mathrm{WL}=\sum_{i=1}^{L-1}\left|x_{i+1}-x_{i}\right|$$
(6)$$\mathrm{RMS}=\sqrt{\frac{1}{L}\sum_{i=1}^{L}x_{i}^{2}}$$
where $x_{i}$ is the value of the signal at the *i*th sample. It is worth noting that, unlike in [39], the WL-PHASOR and RMS-PHASOR were used separately and then tested in combination as in [39], where they are referred to as the PHASOR feature set. This was performed to assess whether each *f*-PHASOR set can provide reliable performance on its own without the need for aggregation into larger, and thus more computationally expensive, feature sets. Moreover, other recent state-of-the-art feature sets that do not employ the phasor representation were used for comparison. In particular, the Hudgins time domain feature set (HTD) [6], time-domain power spectral descriptors (TDPSD) [44], time-domain features with autoregressive coefficients (TDAR) [30], and the feature set proposed in [15] (Du) were used as standard hand-crafted feature sets for comparison. Additionally, the proportion of positive peaks and the maximum peak value of the convolutional layer of Rocket and Mini-Rocket were considered as feature sets and used for classification in accordance with the tools proposed in [40,41].

For each subject and trial, features were extracted using a window length of 150 ms with a sliding increment of 25 ms [4]. The data were used to assess the quality of the feature spaces, then normalized with the z-score approach to perform pattern recognition experiments. All signal processing, feature extraction and pattern recognition tests were conducted using Python 3.11.4 on a computer with a core i7 processor, a RAM of 16 GB, and an NVIDIA GeForce GTX 1060 GPU (Nvidia, Santa Clara, CA, USA).

### 2.3. Feature Space Quality Metrics

In order to assess the quality of the class separability in the feature spaces considered in Section 2.2, two specific metrics used in myoelectric pattern recognition were used [7]. In particular, the first metric employed was the separability index (SI) [45]. The SI can be computed following the procedure in [45]; thus, given the covariance matrix of the data belonging to class *i*, expressed through $\Sigma_{i}$, and the covariance matrix of the data belonging to the most conflicting class $\Sigma_{C_{i}}$, one can first compute the average covariance matrix Σ as
(7)$$\Sigma =\frac{\Sigma_{i}+\Sigma_{C_{i}}}{2}$$

Then, the SI is computed using Σ following the procedure in [45]. Thus,
(8)$$\mathrm{SI}=\frac{1}{K}\sum_{i=1}^{K}\left(\frac{1}{2}{\left\{{\left(m_{i}-m_{C_{i}}\right)}^{T}\Sigma^{-1}\left(m_{i}-m_{C_{i}}\right)\right\}}^{\frac{1}{2}}\right)$$
where *i* indicates the specific cluster of data belonging to the *i*th class. K is the total number of classes considered; thus, in this study, it was set to 5. $m_{i}$ and $m_{C_{i}}$ represent, respectively, the centroid of the *i*th cluster and its most conflicting one [43,45]. The SI mirrors the distance between the classes in the assessed feature space. Hence, the greater the SI, the better the capability of the feature in distinctly mapping each gait phase.

The second metric employed is the mean semi-principal axis (MSA), which quantifies the compactness of the clusters in the feature space [7,45]. To compute this metric, each cluster is approximated as a hyper-ellipsoid in the feature space using singular value decomposition on the data belonging to each class. Then, the geometric mean is applied to the singular values as follows:(9)$$\mathrm{MSA}=\frac{1}{K}\sum_{i=1}^{K}{\left(\prod_{p=1}^{D}a_{ip}\right)}^{\frac{1}{D}}$$
where $a_{ip}$ is the *p*th singular value of the *i*th class, and *D* indicates the dimension of the feature space. MSA reflects the agglomeration properties of each cluster, i.e., it accounts for the inner variance of the clusters; thus, the lower the MSA, the more compact the cluster is in the considered feature space.

### 2.4. Pattern Recognition Models and Testing

Two pattern recognition models used in myoelectric control, i.e., SVM with Radial base function (RBF) kernel, the γ parameter of the kernel set as the product between the number of the features and the variance of the features, and LDA with the principal diagonal covariance matrix model were implemented using Python libEMG toolbox [46] https://libemg.github.io/libemg/index.html (accessed on 8 April 2024), which represents a useful framework for modeling pattern recognition systems [46]. Models were trained intra-subjects using the feature sets described in Section 2.2, and using a five-fold leave-one-trial-out testing scheme (five-fold LOTO). This was performed to obtain a robust estimation of the performance metrics provided by the classifiers. The same approach was used to assess Rocket and Mini-Rocket architectures [40,41]. In this case, the pre-processed EMG windows were used directly to feed both architectures, leaving to them the hidden extraction of possible spatial activation patterns through the large convolutional layer, which was set to 84 kernels both for Rocket and Mini-Rocket.

Two metrics were used to assess the goodness of the architectures, accuracy (ACC) and the Matthews correlation coefficient (MCC) [2]; such metrics guarantee a clearer picture of the classifier performance when the testing data are unbalanced. This is common in gait phase recognition when the number of samples belonging to the stance and swing with their relative subphases are different. Both ACC and MCC were evaluated in the five-fold LOTO, and are presented averaged among the 40 subjects. Moreover, for each subject and each model, the computational time, which includes the extraction of testing features and the classification of data points, was considered a computational performance metric in the analysis. To ensure fairness and reproducibility among all different feature sets and machine learning models, the same computer was used for all computations. Additionally, features were extracted using the libEMG package [46], or the Rocket and Mini-Rocket open-source codes [40,41] to ensure the possibility of replicating the implementation on any other machine.

## 3. Results

### 3.1. Feature Space Quality Metrics

The SI and MSA metrics described in Section 2.3 were computed for each subject and for each feature set described in Section 2.2. Figure 1 reports the mean SI among the 40 subjects contained in the dataset. The PHASOR feature set showed the highest SI value of 7.7 with the greatest standard deviation of 2.1, which is consistently greater than all the other feature sets. Indeed, all the hand-crafted feature sets showed a mean SI not greater than 3.3, which was obtained for the TDAR set. In contrast, Rocket and Mini-Rocket showed mean SI values of 6.5 and 4.3, respectively, indicating superior separability properties compared to the hand-crafted feature sets.

Regarding the mean MSA, the WL-PHASOR showed the lowest value of 0.24, followed by RMS-PHASOR and TDPSD, which respectively showed mean MSA values of 0.35 and 0.44 as shown in Figure 2. Moreover, PHASOR showed a comparable MSA with respect to the aforementioned feature sets, i.e., mean value of 0.5, which suggests the repeatable mapping of myoelectric activity in the five considered gait phases. It is noteworthy that all the other feature sets showed greater mean MSA, especially Rocket, which showed the highest value of 2.5, indicating low compactness of the clusters, and thus more spread patterns in the feature space.

### 3.2. Performance Metrics and Computation Time

The mean ACC of LDA, SVM, Rocket, and Mini-Rocket are reported in Figure 3. It is noteworthy that PHASORS showed the best performance among all the comparisons, with an accuracy greater than 82%, whereas the other classifiers did not exceed 80%. Moreover, a convolutional kernel approach like Rocket did not outperform the majority of the hand-crafted feature sets analyzed when used with SVM, although it performed always better than LDA (see Figure 3). On the other hand, Mini-Rocket achieved an ACC of 56%, making it the poorest model in terms of automatic gait phase recognition. Regarding the comparison between LDA and SVM models, the former performed worse than the latter in each hand-crafted feature set investigated, except for TDPSD, where the results are comparable. However, in this case, the mean ACC obtained was not greater than 76%.

A similar analysis can be carried out using the MCC values reported in Table 1. All the MCC values obtained were positive, indicating that none of the models were completely influenced by randomness in their decision outputs. PHASOR with SVM showed the best MCC, while Rocket provided the lowest. It deserves to be noticed that RMS-PHASOR and WL-PHASOR with SVM also showed values greater than 0.74, indicating good generalization capabilities even in unbalanced data conditions. Furthermore, the MCC values confirm the superiority of SVM compared to LDA (see Table 1). Consistent with the ACC results reported in Figure 3, the Mini-Rocket classifier demonstrated a greater capability of generalizing the data compared to Rocket, but in any case, it showed lower MCC with respect to PHASOR.

The running time analysis reported in Figure 4 highlights the mean time required for the extraction and classification of testing data. It is noteworthy that among the hand-crafted feature sets, PHASORS, RMS-PHASORS, and WL-PHASORS showed computation times that are comparable to state-of-the-art feature sets such as HTD and Du. These feature sets also exhibited lower computation demands compared to TDPSD and TDAR, which required running times greater than 35 ms. The comparison between SVM and LDA suggests that the latter is slightly superior to the former in terms of running time. Furthermore, Rocket and Mini-Rocket demonstrated running times greater than 100 ms, which were consistently higher than those of all the other feature sets and models employed.

## 4. Discussion

Integrating human volitional control through neuromuscular information extracted from EMG signals is pivotal for advancing the current technologies available for lower limb assistance [4,11]. From this perspective, the data processing pipeline and classification architecture employed are crucial components. On one hand, the quality of the feature space ensures a reliable and consistent mapping of muscle electrical activity into distinct, easily separable, and classifiable patterns. On the other hand, the efficiency of the architecture, in terms of reduced computational cost and energy consumption, is essential for the practical implementation of machine learning systems in portable devices. This suggests the use of well-established and shallow machine learning models, such as SVM and LDA, as employed in this study. Hence, one can focus on enhancing the feature extraction process to capture spatial activation patterns for improving class separability [2].

The analysis performed using the SI and MSA metrics highlights that WS-PHASOR, RMS-PHASOR, and PHASOR can be more robust compared to other classical feature sets like HTD, Du, TDPSD, and TDAR, which do not employ a phasor approach to highlight spatial synergy patterns in EMG signals. In particular, as highlighted in Section 3.1, the PHASOR feature space showed the greatest mean SI among the 40 subjects present in the data, confirming that such a feature set consistently separates cluster centroids, thus making gait phases more distinguishable. Additionally, MSA suggested the significant compactness of the data clusters in the PHASOR feature space, indicating a low level of interference between two contiguous clusters. This is important to reduce the false positive or false negative detection of the actual gait phase, thus protecting the user from undesired transitions and avoiding excessive metabolic costs. The highlighted separability was mirrored by the ACC of SVM models compared to LDA. This may be attributed to two aspects: first, phasor-based feature extraction schemes tend to enlarge the feature space, which naturally fits with the SVM characteristics, i.e., they work well in large dimensional spaces. Secondly, LDA works better with Gaussian features, whereas the SVM approach is geometric and can better deal with non-linear patterns [47]. To further support the effectiveness of the best model, i.e., SVM trained with PHASOR, Figure 5 shows the confusion matrices obtained during testing, averaged among the subjects. It is evident that convolutional kernel approaches such as Rocket and Mini-Rocket had a higher fault detection rate as indicated by the less clean confusion charts. In contrast, PHASOR demonstrated a high capability in reducing the misclassification rate.

The MCC analysis supports the ACC results obtained in testing as reported in Table 1. Given the intrinsically unbalanced nature of the data, i.e., the number of samples attributed to each gait phase is naturally different [21], a positive MCC greater than 0.75, as obtained with PHASOR and SVM, demonstrates the robustness of the model even under unbalanced data conditions. Furthermore, the computational time obtained with PHASOR is not significantly different from HTD, which is fast due to containing only time-domain features (see Figure 3). Moreover, the running time analysis shows that TDAR and TDPSD seemed to require more computational time with respect to the other sets. Although this can be expected for TDAR, what was observed for TDPSD seems to contradict past works [34,44]. However, this can be imputed to the libEMG implementation, which recomputes the three even spectral moments for each moment-dependent feature, unavoidably expanding the computational time.

It deserves to be noted that in this study, a wide range of myoelectric feature extraction schemes were tested on the SIAT-LLMD dataset, which were not assessed in previous works [21] This ensured that the f-PHASOR extraction approach was evaluated against consolidated approaches that were not yet used in lower limb gait phase recognition, making the comparisons fair and highlighting the possible limitations of standard approaches that do not encompass synergistic effects in myoelectric activations. Indeed, typical feature sets, including TDAR and TDPSD, which are among the most suitable for myoelectric control of upper limb devices [30], produced lower performance results in the five-phase detection problem considered in this study. This may be attributed to the fact that, in gait, more than other motor tasks, the extraction of features that carry synergistic myoelectric patterns related to spatial information can be essential to unfold the intrinsic complex nature of decoding volitional neuromuscular control [39,48]. However, although the aforementioned aspect was exploited for problems involving the upper limb [39,48], it seems to be less encountered for lower limb assistive devices, even if the recent literature highlights the importance of using myoelectric information to decode myoelectric patterns for smoothing the interaction between human and exoskeleton or prostheses in the lower limb [12,49]. The present results confirm that new feature extraction schemes based on the proposed approach can enhance the detection accuracy of standard machine learning models embedded in microcontrollers for gait phase recognition. In practical scenarios, such as prosthetic control, this represents a useful result since the f-PHASOR can benefit from embedding the positional information of electrode locations available in the prosthetic socket. These electrodes are generally located radially with respect to the residual limb and thus are easy to associate with the f-PHASOR extraction method, which was generalized in this study to N electrodes. On the other hand, no particular benefits were observed when fully data-driven convolutional transform approaches, such as Rocket or Mini-Rocket, were applied. This was confirmed by the poor ACC and MCC obtained for Rocket, while relatively good performance results were achieved with Mini-Rocket, suggesting that the Rocket architecture was prone to overfitting the data. In any case, the running time analysis indicated higher computational demands for such architectures compared to LDA and SVM, revealing the bottleneck of deep learning methodologies when tested in a five-fold LOTO scheme, i.e., Rocket and Mini-Rocket produce results comparable with standard machine learning approaches, supporting the importance of smart synergistic muscle activation patterns as highlighted by f-PHASOR.

Although a direct comparison with other works was beyond the aim of this study, given the absence of works that used the same dataset and the LOTO approach, it can be observed that the ACC obtained with PHASOR, with a granularity of five phases, is comparable to the ACC obtained in [22,23], showing accuracy greater than 85% under optimal conditions. It should also be noted that the type of population selected for testing the methodology, i.e., healthy subjects, is in line with other studies dealing with the development of myoelectric interfaces for lower limb prosthetic control [31,32]. Moreover, the number of subjects in previous studies was consistently lower, i.e., not greater than 10 [22,23], and their testing procedures did not account for the LOTO scheme, which is less commonly employed but more useful for accurately estimating the actual performance of the trained models. The high variability of the accuracy in [22,23] may be related to the use of standard myoelectric feature sets, which were challenged in correctly discriminating among the different phases when the granularity of the gait cycle was increased [50]. This motivated the introduction of electroencephalographic or mechanical information to improve gait phase detection performance and reduce classifier performance variability while maintaining a consistent granularity of the gait cycle equal to or greater than five [50,51]. However, the use of PHASOR seems to enhance performance without the need for additional sensors attached to the body of the subject, relying only on the spatial EMG setup configuration on the leg, making this approach valuable for further investigation. It is worth mentioning that although the f-PHASOR was tested on a large number of subjects, i.e., 40, none of them had amputations. Therefore, the direct applicability of the methodology to a real-case scenario of prosthetic control was not tested. This constitutes a limitation of the present work, which can, however, be addressed in future studies. Indeed, the f-PHASOR approach can also be applied with EMG signals recorded through the EMG channels located in the prosthetic socket [52]. In this case, although the f-PHASOR necessarily works with myoelectric information recorded from the electrodes radially located in the socket, it should be able to highlight the synergistic effects of physically close muscle fibers. This idea of transitioning from sparse setups, i.e., considering multiple muscle locations [4], to dense setups, i.e., collecting the myoelectric activity of closely related muscle fibers, is not surprising, and it was used in upper limb hand gesture recognition from amputees with good applicability as demonstrated in [53]. This approach can also be transferred to lower limb prosthetic sockets [52], which will allow the applicability of f-PHASOR by computing features with high synergistic information content and thus with a greater separability power as shown in this study.

Ultimately, the classification performance results reported in this study were not altered by any kind of postprocessor, which generally processes the classification output stream by smoothing the decision output and ensuring feasible actuation at the lower level of specific control policies of the active elements belonging to the assistive device. Typical examples of such postprocessors are based on majority voting or Bayesian approaches as widely described in the literature [2,33]. Thus, it is important to note that when dealing with the real-time implementation of the architecture proposed in this work, it can be particularly beneficial to include a postprocessing scheme that mitigates possible spurious misclassifications by refining the decision output using a queue of past classifier states along with the current one. This aspect can play a crucial role in avoiding unwanted behavior of the myoelectric interface, thereby increasing the safety of the user [2].

## 5. Conclusions

In this paper, an *f*-PHASOR approach was proposed as a new way to embed spatial muscle synergy information into the feature space to enhance gait phase recognition without additional information. This approach can be useful for upgrading the performance of shallow classifiers used in lower-limb myoelectric control applications. Indeed, the feature set separability metrics, together with the accuracy performance obtained in five-fold LOTO testing, suggest the superiority of the proposed approach compared to state-of-the-art feature sets used for myoelectric control. 

## Figures and Tables

**Figure 1 sensors-24-05828-f001:**
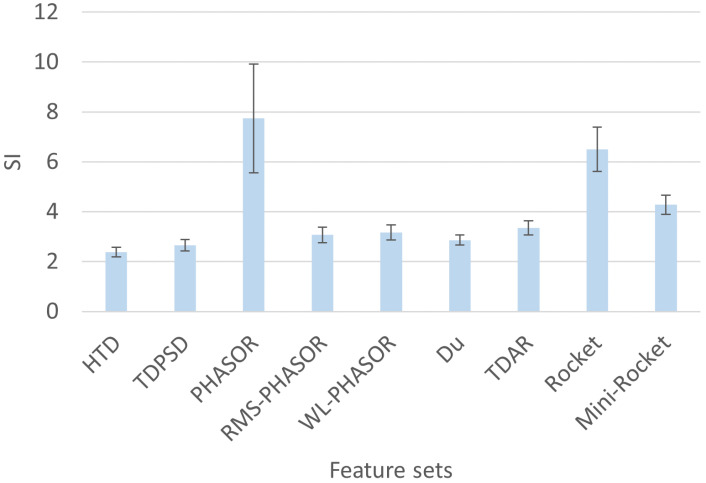
Mean SI obtained in testing for the 40 subjects analyzed. PHASOR feature set obtained the best performance when used with SVM among all the feature sets and models employed.

**Figure 2 sensors-24-05828-f002:**
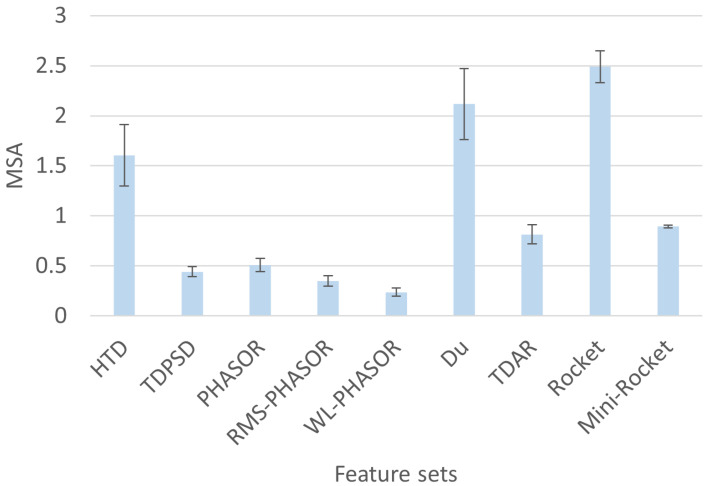
Mean MSI obtained in testing for the 40 subjects analyzed. PHASOR feature set obtained the best performance when used with RBF-SVM among all the feature sets and models employed.

**Figure 3 sensors-24-05828-f003:**
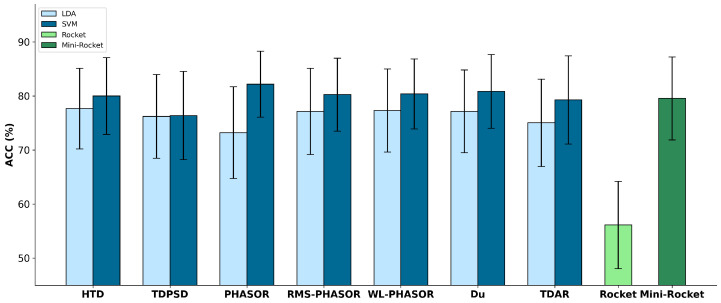
Mean accuracy (ACC) obtained in testing for the 40 subjects analyzed. PHASOR feature set obtained the best performance when used with SVM among all the feature sets and models employed.

**Figure 4 sensors-24-05828-f004:**
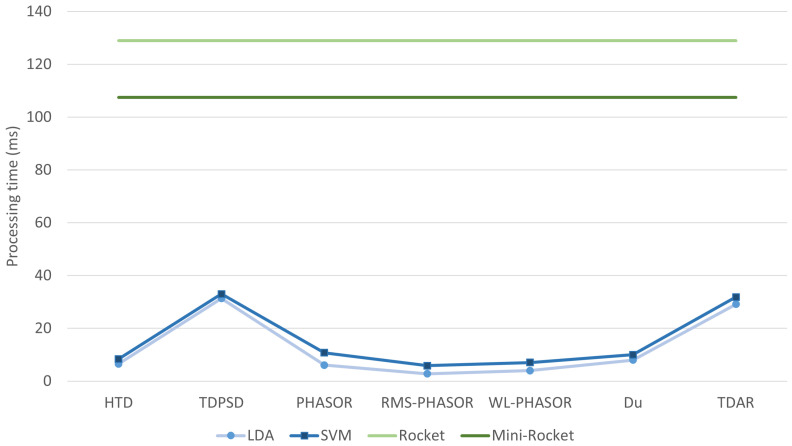
Mean processing time in ms for computing the feature sets in testing and performing the classification output. PHASORS, RMS-PHASORS and WL-PHASORS provided processing times comparable with other hand-crafted feature sets as HTD and Du, and showed better computational performance with respect to TDPSD and TDAR sets, for both LDA and SVM classifiers. Rocket and Mini-Rocket showed a consistently higher computational demand with respect to the hand-crafted features.

**Figure 5 sensors-24-05828-f005:**
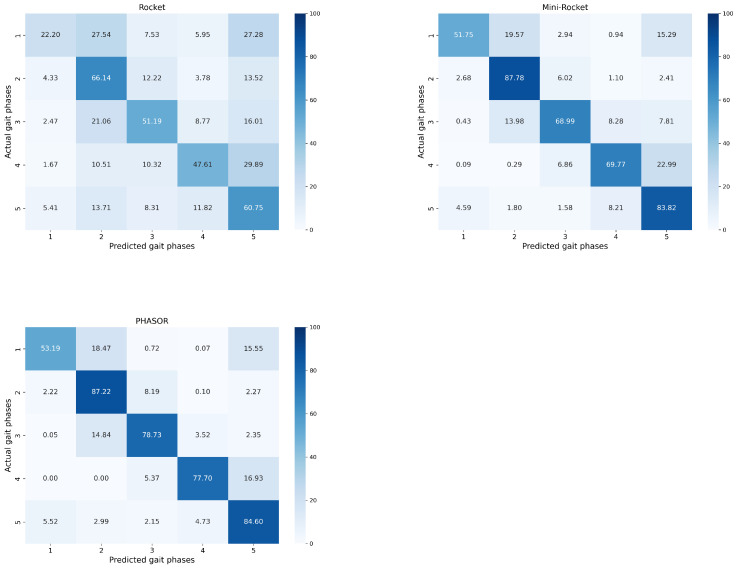
Confusion matrices for Rocket, Mini-Rocket, and PHASOR averaged among subjects. PHASOR exhibits a dominant principal diagonal with low misclassification rates outside the principal diagonal. In contrast, Rocket shows the worst performance, while Mini-Rocket performs well but not as well as PHASOR. Overall, the confusion matrices confirm the analysis performed using SI and MSA metrics.

**Table 1 sensors-24-05828-t001:** The Matthews correlation coefficient (MCC) obtained in testing averaged among the 40 subjects analyzed as reported in Section 2.2. The MCC ranges from −1 to 1. Values close to 1 indicate very good prediction, values equal to 0 indicate random prediction by the classifier, while values that tend towards negative indicate consistent mistakes in the classifier output. PHASOR with SVM obtained the highest MCC as highlighted by bold numbers, confirming the accuracy trend shown in Figure 3.

Classifier	HTD	TDPSD	PHASOR	RMS-PHASOR	WL-PHASOR	Du	TDAR
SVM	0.74 ± 0.10	0.69 ± 0.12	**0.77 ± 0.08**	0.75 ± 0.10	0.74 ± 0.09	0.75 ± 0.10	0.71 ± 0.13
LDA	0.71 ± 0.11	0.69 ± 0.11	0.64 ± 0.13	0.70 ± 0.12	0.71 ± 0.11	0.71 ± 0.11	0.67 ± 0.12
Rocket	0.36 ± 0.14						
Mini-Rocket	0.74 ± 0.10						

## Data Availability

Data are contained within the article.

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
