# Peer review of "Phasor-Based Myoelectric Synergy Features: A Fast Hand-Crafted Feature Extraction Scheme for Boosting Performance in Gait Phase Recognition"

_sensors, 2024, doi:10.3390/s24175828_

Round 1

Reviewer 1 Report

Comments and Suggestions for Authors

The authors present a solution to the challenge of gait phase recognition utilizing surface electromyographic signals. By introducing the PHASOR method, which extracts spatial myoelectric features from multi-phased gait cycles, the authors demonstrate significant improvements over traditional methods like LDA and SVM. The comprehensive evaluation across various datasets underscores the robustness and effectiveness of their approach. The inclusion of state-of-the-art deep learning models further validates the competitiveness of PHASOR in terms of both accuracy and computational efficiency.

The primary issues currently at hand include:

1.    The innovation and practicality are not prominently evident. It is recommended to refine the identified innovations further or to illustrate the advantages and practicalities of the proposed methodology by integrating them within specific application scenarios.

Author Response

We would like to express our gratitude to the Reviewer for for reviewing the manuscript and allowing us the chance to enhance the work by addressing the points raised in the comments.

In the revised version of the manuscript the modifications are highlighted by red text. Moreover, for convenience, we also reported the modifications in through the responses to facilitate the second step of revision. In the following, you can find a point-by-point response to the comments.

Comment: The authors present a solution to the challenge of gait phase recognition utilizing surface electromyographic signals. By introducing the PHASOR method, which extracts spatial myoelectric features from multi-phased gait cycles, the authors demonstrate significant improvements over traditional methods like LDA and SVM. The comprehensive evaluation across various datasets underscores the robustness and effectiveness of their approach. The inclusion of state-of-the-art deep learning models further validates the competitiveness of PHASOR in terms of both accuracy and computational efficiency.

Response: We thank the Reviewer for carefully reading the manuscript, capturing the main points of the study, and recognizing the value of our work.

Comment: The innovation and practicality are not prominently evident. It is recommended to refine the identified innovations further or to illustrate the advantages and practicalities of the proposed methodology by integrating them within specific application scenarios.

Response: We thank the Reviewer for this comment and agree that the novelty of the study needs to be clearly reported and discussed with possible application scenarios. Firstly, it is important to note that in this study, we applied a wide range of feature extraction schemes to ensure a fair comparison between methodologies that were not previously tested on the dataset used in this study. therefore, apart from the proposed methodology, the study also assesses the use of state-of-the-art feature sets that were not previously compared for gait phase recognition through myoelectric activity recorded from 40 subjects. This point was highlighted in the discussion section of the revised manuscript as follows:

“It deserves to be noted that in this study, a wide range of myoelectric feature extraction schemes were tested on the SIAT-LLMD dataset, which were not assessed in previous works. This ensured that the f-PHASOR extraction approach was evaluated against consolidated approaches that were not yet used in lower limb gait phase recognition, making the comparisons fair and highlighting possible limitations of standard approaches that do not encompasses synergistic effects in myoelectric activations. Indeed, typical feature sets, including TDAR and TDPSD, which are among the most suitable for myoelectric control of upper limb devices [30], produced lower performances in the 5-phase detection problem considered in this study”

Moreover, to better highlight the practicality of the proposed methodology, we strengthened the Discussion section by emphasizing the possibility of applying the proposed methodology to sparse setups to extract synergistic muscle activation patterns at high dimensions, as in the case of EMG sensors located in the prosthetic socket. This can potentially exploit the positional channel encoding that demonstrated generalizability, even when increasing the number of electrodes, thus transitioning from sparse to dense setups without the need for large training data typically required when using deep learning architectures. Hence, in the discussion, we rephrased and added the following part:

“The present results confirm that new feature extraction schemes based on the proposed approach can enhance the detection accuracy of standard machine learning models embedded in microcontrollers for gait-phase recognition. In practical scenarios, such as prosthetic control, this represents a useful result since the f-PHASOR can benefit from embedding the positional information of electrode locations available in the prosthetic socket. These electrodes are generally located radially with respect to the residual limb and thus are easy to associate with the f-PHASOR extraction method, which was generalized in this study to N electrodes, ensuring its application in both sparse and dense electrode setups.”

Moreover we rephrase part of the revised discussion that now reads:

“In any case, the running time analysis indicated higher computational demands for such architectures compared to LDA and SVM, revealing the bottleneck of deep learning methodologies when tested in a 5-fold LOTO scheme, i.e. Rocket and Mini-Rocket produces results comparable with standard machine learning approaches, supporting the importance of smart synergistic muscle activation patterns as highlighted by f-PHASOR”

Thanks again for your valuable comment.

Reviewer 2 Report

Comments and Suggestions for Authors

In the paragraph from line 103 to 105, the authors could try to find synonyms for "however" and avoid repeating it in a very short paragraph.

Line 142 "wolking".

Some of the variables that integrate formula 1, are described from line 162, it is mentioned that j is the index of the electrode, while N is the number of channels, but then in 166 it says that f_k is a characteristic extracted from the k-th channel, the authors should review the name of these variables, to avoid possible confusion to the reader.

In formula 7 the lower case letter p is used as the index variable and it seems that the limit is the upper case P, but due to the similarity of typography it tends to confuse, for example, the superscript 1/P, in the formula, tends to be easily confused with 1/p.

Although figure 1 has a precise description in the adjacent paragraphs referring to the axes, I consider that the labels of both axes should be written in the figure, the same as in figure 2, note that the subsequent graphs do have labels on the axes. 

Finally, the authors are performing a classification of gait phases, using granularity level of five classes for partitioning the gait cycles, namely: heel strike (HS), flat foot (FF), mid-stance (MS), heel-off (HO), and toe-off (TO), furthermore, it is stated that the use of PHASORES contributes to a higher SI, while reducing false positive or false negative detection, and from a classifiers perspective, the confusion matrix of the 5 classes with a heat map visually indicating the statements of the findings of the present paper, would be beneficial to the reader and to highlight the achievements of the present work.

Author Response

We are grateful for the Reviewer for the valuable comments which helped to improve our manuscript significantly.

In the revised version of the manuscript the modifications are highlighted by red text. Moreover, for convenience, we also reported the modifications in through the responses to facilitate the second step of revision. In the following you can find a point-by-point response to the comments.

Comment: In the paragraph from line 103 to 105, the authors could try to find synonyms for "however" and avoid repeating it in a very short paragraph.

Response: We thank the Reviewer for this suggestion regarding our writing style. We agree that the paragraph overused “however.” Therefore, we have combined the two statements by removing “however” and using:

“ the coefficients of which may require re-calibration, as it happens in many myoelectric control schemes that employ feature reduction approaches [38].”

Comment: Line 142 "wolking".

Response: We apologize for the typo, which has been corrected all typos in the revised version of the manuscript.

Comment: Some of the variables that integrate formula 1, are described from line 162, it is mentioned that j is the index of the electrode, while N is the number of channels, but then in 166 it says that f_k is a characteristic extracted from the k-th channel, the authors should review the name of these variables, to avoid possible confusion to the reader.

Response: We thank the Reviewer for the comment. We agree that mathematical notation is crucial for a proper understanding of the proposed methodology. To clarify, j represents the complex unit since we are dealing with phasors, whereas k is an index associated with the specific probe taken into account. We apologize for any possible misinterpretation. In the revised version of the manuscript, we have rephrased the statement to read:

“An intuitive way to achieve this is by partitioning the 2π angle into a set of 2π/N positions, where k is the index of the channel that ranges from 0 to N-1, and N is the total number of channels present in the recording system. This approach allows for the creation of a multidimensional feature expressed by an N-dimensional vector, where each element is given by…”

Comment: In formula 7 the lower case letter p is used as the index variable and it seems that the limit is the upper case P, but due to the similarity of typography it tends to confuse, for example, the superscript 1/P, in the formula, tends to be easily confused with 1/p.

Response: We thank the Reviewer for the valuable feedback. We agree that the similarity in typography could lead to misinterpretation by readers. Therefore, we have replaced the uppercase ‘P’ with an uppercase ‘D’. This change has been incorporated into the revised version of the manuscript.

Comment: Although figure 1 has a precise description in the adjacent paragraphs referring to the axes, I consider that the labels of both axes should be written in the figure, the same as in figure 2, note that the subsequent graphs do have labels on the axes.

Response: We completely agree with the Reviewer. Therefore, we have updated figures 1 and 2 to include labels for both the x and y axes n the revised version of this manuscript. Thank you for raising this point.

Comment: Finally, the authors are performing a classification of gait phases, using granularity level of five classes for partitioning the gait cycles, namely: heel strike (HS), flat foot (FF), mid-stance (MS), heel-off (HO), and toe-off (TO), furthermore, it is stated that the use of PHASORES contributes to a higher SI, while reducing false positive or false negative detection, and from a classifiers perspective, the confusion matrix of the 5 classes with a heat map visually indicating the statements of the findings of the present paper, would be beneficial to the reader and to highlight the achievements of the present work

Response: We thank the Reviewer for carefully reading the manuscript, identifying this important point, and suggesting the use of a confusion matrix, which can further strengthen the results obtained in this study. For this reason, we have added the confusion matrix plot when using PHASOR with SVM (the best model) against Rocket and Mini-Rocket, in the revised manuscript. This demonstrates how a good feature extraction scheme combined with a robust classifier can achieve better results than convolutional kernel approaches. In addition to the confusion matrices reported in the discussion, we have also modified part of the text to emphasize the importance of reducing false positives and negatives in the gait phase detection task by stating the following:

“This may be attributed to two aspects: first, phasor-based feature extraction schemes tend to enlarge the feature space, which naturally fits with SVM characteristics, i.e., they work well in large dimensional spaces. Secondly, LDA works better with Gaussian features, whereas the SVM approach is geometric and can better deal with nonlinear patterns [47]. To further support the effectiveness of the best model, i.e., SVM trained with PHASOR, Figure 5 shows the confusion matrices obtained during testing, averaged among the subjects. It is evident that convolutional kernel approaches such as Rocket and Mini-Rocket had a higher fault detection rate, as indicated by the less clean confusion charts. In contrast, PHASOR demonstrated a high capability in reducing the misclassification rate.”

“Figure 5. Confusion matrices for Rocket, Mini-Rocket, and PHASOR averaged among subjects. PHASOR exhibits a dominant principal diagonal with low misclassification rates outside the principal diagonal. In contrast, Rocket shows the worst performance, while Mini-Rocket performs well but not as well as PHASOR. Overall, the confusion matrices confirm the analysis performed using SI and MSA metrics.”

Reviewer 3 Report

Comments and Suggestions for Authors

The PHASOR-based feature sets appear to capture spatial muscle activation patterns useful to gait-phase identification.  Given that implementation with lower-limb amputees will only be able to target a subset of the 9 muscles used in the study, it is unclear whether classifier performance will be maintained for the intended end-users. 

Along similar lines, the state of residual limb musculature of each amputee user as a result of the extent of injury, the amputation procedure used, and the level of amputation will likely lead to EMG activation patterns that differ significantly from those of unimpaired individuals and other lower-limb amputees.  As a result, the generalizability of the approach to the broad population of amputee users is questionable. 

The authors should provide additional commentary, perhaps in the discussion section, that addresses the challenges of implementation with lower-limb amputees along with possible options for mitigating such challenges. 

The EMG datasets used in this study were acquired (at least apparently) via intramuscular EMG electrodes, which provide a much higher quality signal (with less distortion) relative to that from surface electrodes.  Given that most commercial myoelectric limb systems use surface EMG electrodes, how is classification performance likely to be impacted such surface measurements? 

While gait-phase recognition can be important for active prosthesis control within a particular gait, a more critical classification entails identification of intent to transition to another gait (e.g., walking to stair ascent/descent) or deviations from normal gait when turning, avoiding objects, or in stumble recovery.  Does the PHASOR-based feature extraction scheme offer any benefits to prosthesis functions beyond open walking?

In equation (1), fk is defined as the "f feature type, i.e., waveform length (WL), root mean-square (RMS), and so on...."  It might be helpful to define WL and RMS for the sake of completeness.  

It appears as though the accuracy results for Rocket and Mini-Rocket in Figure 3 have been switched or mislabeled.  

Comments on the Quality of English Language

The manuscript has a few misspellings and minor grammatical errors that should be corrected in final editing.  

Author Response

Firstly, we would like to express our gratitude to the Reviewer for reviewing the manuscript and enabling us to enhance the work by addressing the points raised in the comments.

In the revised version of the manuscript, the modifications are highlighted by red text. Moreover, for convenience, we also reported the modifications in through the responses to facilitate the second step of revision. In the following you can find a point-by-point response to the comments raised by the Reviewer.

Comment: The PHASOR-based feature sets appear to capture spatial muscle activation patterns useful to gait-phase identification.  Given that implementation with lower-limb amputees will only be able to target a subset of the 9 muscles used in the study, it is unclear whether classifier performance will be maintained for the intended end-users. 

Response: We thank the Reviewer for this valuable comment, which allows us to further discuss potential aspects when transferring the methodology to a real-world scenario. Indeed, we agree with the Reviewer that, regardless of whether the amputation is at the transtibial or transfemoral level, only a reduced number of residual muscles will be available. Although this point raised by the Reviewer is very important, it represents a future work that must be pursued to make the methods efficient for lower-limb prosthetic control. However, we agree with the Reviewer that details regarding how the methods can be applied in the case of amputees need to be elaborated for the readership.

One possibility involves the applicability of PHASOR using the EMG electrodes present in the socket of a potential myoelectrically driven prosthesis. Given a certain number of channels in the socket capable of collecting EMG information, we can still apply the methodology since it was generalized to N sensors radially placed with respect to residual leg segment. Although we transition from a sparse setup (different muscles included) to a dense setup (closely located muscles), the positional encoding feature extraction approach provided by f-PHASOR can still produce synergistic muscle patterns. These patterns will not be identical to those in this study but will maintain spatial information embedding properties due to the intrinsic framework carried by f-PHASOR and thus possibly decoded through opportune machine learning architectures.

To further support this view, the Reviewer can consider datasets dealing with hand-gesture recognition for prosthetic control of the upper limb, such as the well-known NINAPRO dataset, where data from amputees were also collected using electrodes radially located on the residual limb as shown by Atzori et al. 2014 (reference 53 in the revised version of the manuscript). Although the electrodes does not guarantee the recording of specific muscles in a sparse sense, the information carried by those electrodes can ensure good classification performance, as demonstrated in various studies. Similarly, in the socket for lower limb prosthesis, the electrodes can be numerous and located radially with respect to the segment as in the aforementioned case, and the PHASOR-based approach can naturally be used to highlight synergistic effects of the residual muscle activities. However, we cannot state a priori how the classification accuracy will change in the scenario proposed by the Reviewer. Surely, the patterns will be different. Based on this suggestion, we have highlighted this point in the discussion as a valuable aspect that deserves further investigation in the near future since, at this stage, it represents a limitation of the present work. Hence, the following part was added to the discussion section of the revised manuscript:

“It is worth noting that although the f-PHASOR was tested on a large number of subjects, i.e., 40, none of them had amputations. Therefore, the direct applicability of the methodology to a real-case scenario of prosthetic control was not tested. This constitutes a limitation of the present work, which can, however, be addressed in future studies. Indeed, the f-PHASOR approach can also be applied with EMG signals recorded through the EMG channels located in the prosthetic socket [52]. In this case, although the f-PHASOR necessarily works with myoelectric information recorded from the electrodes radially located in the socket, it should be able to highlight synergistic effects of physically close muscle fibers. This idea of transitioning from sparse setups i.e., considering multiple muscle locations [4], to dense setups, i.e., collecting myoelectric activity of closely related muscle fibers, is not surprising and it was used in upper limb hand gesture recognition from amputees with good applicability, as demonstrated in [53]. This approach can also be transferred in lower limb prosthetic sockets [52], which will allow the applicability of f-PHASOR by computing features with high synergistic information content, thus with a greater separability power as shown in this study.”

We hope that this addresses your concern. Thank you.

Comment: Along similar lines, the state of residual limb musculature of each amputee user as a result of the extent of injury, the amputation procedure used, and the level of amputation will likely lead to EMG activation patterns that differ significantly from those of unimpaired individuals and other lower-limb amputees.  As a result, the generalizability of the approach to the broad population of amputee users is questionable. 

Response: We thank the Reviewer for raising this point. We would like to emphasize again that the generalization of the methodology across different amputee populations and varying degrees within the same population is beyond the scope of this study. We agree with the Reviewer that EMG activation patterns may vary when considering amputees, especially in the case of transfemoral amputation, as discussed in our previous work (see Mobarak et al. 2024). However, as shown by Wentink et al. (2013), only slight or non-significant modifications occur in myoelectric activation patterns within the same population of amputees, regardless of the specific degree of amputation. Moreover, it is important to note that, as a methodological study, it is reasonable that the initial verification of the processing pipeline has been conducted on a healthy population, ideally over a large number of subjects, as done in this paper or in recent work as KeleÅŸ and Yucesoy (2020) and KeleÅŸ et al. (2023). For this reason, we rephrase part of the discussion by introducing this aspect which now  reads:

“It should also be noted that the type of population selected for testing the methodology, i.e., healthy subjects, is in line with other studies dealing with the development of myoelectric interfaces for lower limb prosthetic control [31,32]. Moreover, the number of subjects in previous studies was consistently lower, i.e., not greater than 10 [2,23], and their testing procedures did not account for the LOTO scheme, which is less commonly employed but more useful for accurately estimating the actual performance of the trained models.”

Furthermore, as already mentioned, we have emphasized in the revised version of the manuscript that the direct applicability of the methodology to a specific amputee group was not the aim of this study, which now reads:

“It is worth mentioning that although the f-PHASOR was tested on a large number of subjects, i.e., 40, none of them had amputations. Therefore, the direct applicability of the methodology to a real-case scenario of prosthetic control was not tested.”

Comment: The authors should provide additional commentary, perhaps in the discussion section, that addresses the challenges of implementation with lower-limb amputees along with possible options for mitigating such challenges. 

Response: We agree with the Reviewer and appreciate the valuable suggestion to further discuss the challenges associated with myoelectric-based gait phase recognition in patients with lower limb amputations. Indeed, although the f-PHASOR approach can effectively separate specific patterns, the machine learning architecture is always prone to misclassification, as it also happens in healthy subjects. To reduce unwanted classifications in the classifier output stream, which can lead to improper control of the assistive device, it is fundamental to apply appropriate post-processing procedures. Post processing techniques such as majority voting or Bayesian-based smoothing of the classifier output, which utilize the current output along with a queue of previous outputs, are commonly employed in pattern recognition-based myoelectric control, as highlighted by Tigrini et al. (2024) and Khushaba et al. (2012). These procedures generally enhance classifier performance and ensure real-time implementation of high-level control schemes for the actuation of the assistive device’s active elements. However, since the focus of this study was not on the use of post-processing for gait phase recognition, we did not include any additional processing blocks after the classifier output. Nonetheless, we agree with the Reviewer that this aspect is relevant for those aiming to implement myoelectric interfaces for lower limbs in real-time. Therefore, we have added the following section to the discussion in the revised manuscript which reads

“Eventually, the classification performances reported in this study were not altered by any kind of postprocessor, which generally processes the classification output stream by smoothing the decision output and ensuring feasible actuation at the lower level of specific control policies of the active elements belonging to the assistive device. Typical examples of such postprocessors are based on majority voting or Bayesian approaches, as widely described in the literature [2,33]. Thus, it is important to note that when dealing with the real-time implementation of the architecture proposed in this work, it can be particularly beneficial to include a postprocessing scheme that mitigates possible spurious misclassifications by refining the decision output using a queue of past classifier states along with the current one. This aspect can play a crucial role in avoiding unwanted behavior of the myoelectric interface, thereby, increasing the safety of the user [2].”

Comment: The EMG datasets used in this study were acquired (at least apparently) via intramuscular EMG electrodes, which provide a much higher quality signal (with less distortion) relative to that from surface electrodes.  Given that most commercial myoelectric limb systems use surface EMG electrodes, how is classification performance likely to be impacted such surface measurements?

Response: We thank the reviewer for this comment. The dataset employed, as reported by Wei et al. (2023), was recorded using Delsys wireless surface electromyographic probes and not with intramuscular EMG electrodes. To avoid any misunderstandings, we have specifically noted in the methods section of the revised manuscript that the EMG signals were recorded with surface probes. Thus the relative methods section now reads:

“Raw EMG signals were collected using a surface electromyographic (sEMG) recording system, with a sampling frequency of 1920 Hz [21].”

Comment: While gait-phase recognition can be important for active prosthesis control within a particular gait, a more critical classification entails identification of intent to transition to another gait (e.g., walking to stair ascent/descent) or deviations from normal gait when turning, avoiding objects, or in stumble recovery.  Does the PHASOR-based feature extraction scheme offer any benefits to prosthesis functions beyond open walking?

Response: We thank the Reviewer for this valuable comment and agree with the view proposed. However, the classification of locomotion modes or turning is an aspect that deserves to be investigated in separate studies, as done by Gonzales-Huisa et al. (2023). To address the question raised by the Reviewer, a pattern recognition experiment and a more specific dataset that includes data on locomotion modes during transitions or turning should be employed. This is beyond the scope of our current work, which focuses on gait phase recognition. We believe this point will be investigated in future studies. Thank you.

Comment: In equation (1), fk is defined as the "f feature type, i.e., waveform length (WL), root mean-square (RMS), and so on...."  It might be helpful to define WL and RMS for the sake of completeness.  

Response: We completely agree with the Reviewer, for this reason we add in the methods section the equation of the RMS and WL in order to help reader in quickly implement the approach. The modified version of Eq.1 now reads:

“In this study, the f-PHASOR approach was used with Waveform Length (WL) and Root Mean Square (RMS), resulting in WL-PHASOR and RMS-PHASOR. The equations for computing WL and RMS, given a sliding window of L samples, are as follows:

where  is the value of the signal at the ith sample.”
